# Beyond normality: Learning sparse probabilistic graphical models in the non-Gaussian setting

**Rebecca E. Morrison**
MIT
rmorriso@mit.edu

**Ricardo Baptista**
MIT
rsb@mit.edu

**Youssef Marzouk**
MIT
ymarz@mit.edu

## Abstract

We present an algorithm to identify sparse dependence structure in continuous and non-Gaussian probability distributions, given a corresponding set of data. The conditional independence structure of an arbitrary distribution can be represented as an undirected graph (or Markov random field), but most algorithms for learning this structure are restricted to the discrete or Gaussian cases. Our new approach allows for more realistic and accurate descriptions of the distribution in question, and in turn better estimates of its sparse Markov structure. Sparsity in the graph is of interest as it can accelerate inference, improve sampling methods, and reveal important dependencies between variables. The algorithm relies on exploiting the connection between the sparsity of the graph and the sparsity of transport maps, which deterministically couple one probability measure to another.

## 1 Undirected probabilistic graphical models

Given $n$ samples from the joint probability distribution of some random variables $X_1, \ldots, X_p$, a common goal is to discover the underlying Markov random field. This field is specified by an undirected graph $G$, comprising the set of vertices $V = \{1, \ldots, p\}$ and the set of edges $E$. The edge set $E$ encodes the conditional independence structure of the distribution, i.e., $e_{jk} \notin E \iff X_j \perp\!\!\!\perp X_k \mid \boldsymbol{X}_{V \setminus \{jk\}}$. Finding the edges of the graph is useful for several reasons: knowledge of conditional independence relations can accelerate inference and improve sampling methods, as well as illuminate structure underlying the process that generated the data samples. This problem—identifying an undirected graph given samples—is quite well studied for Gaussian or discrete distributions. In the Gaussian setting, the inverse covariance, or precision, matrix precisely encodes the sparsity of the graph. That is, a zero appears in the $jk$-th entry of the precision if and only if variables $X_j$ and $X_k$ are conditionally independent given the rest. Estimation of the support of the precision matrix is often achieved using a maximum likelihood estimate with $\ell_1$ penalties. Coordinate descent (glasso) [4] and neighborhood selection [14] algorithms can be consistent for sparse recovery with few samples, i.e., $p > n$. In the discrete setting, [12] showed that for some particular graph structure, the support of a generalized covariance matrix encodes the conditional independence structure of the graph, while [21] employed sparse $\ell_1$-penalized logistic regression to identify Ising Markov random fields.

Many physical processes, however, generate data that are continuous but non-Gaussian. One example is satellite images of cloud cover formation, which may greatly impact weather conditions and climate [25, 20]. Other examples include biological processes such as bacteria growth [5], heartbeat behavior [19], and transport in biological tissues [9]. Normality assumptions about the data may prevent the detection of important conditional dependencies. Some approaches have allowed for non-Gaussianity, such as the nonparanormal approach of [11, 10], which uses copula functions to estimate a joint non-Gaussian density while preserving conditional independence. However, this approach is still restricted by the choice of copula function, and as far as we know, no fully general approach is yet available. Our goal in this work is to *consistently* estimate graph structure when

the underlying data-generating process is non-Gaussian. To do so, we propose the algorithm SING (Sparsity Identification in Non-Gaussian distributions). SING uses the framework of transport maps to characterize arbitrary continuous distributions, as described in §3. Our representations of transport maps employ polynomial expansions of degree $\beta$. Setting $\beta = 1$ (i.e., linear maps) is equivalent to assuming that the data are well approximated by a multivariate Gaussian. With $\beta > 1$, SING acts as a generalization of Gaussian-based algorithms by allowing for arbitrarily rich parameterizations of the underlying data-generating distribution, without additional assumptions on its structure or class.

## 2 Learning conditional independence

Let $X_1, \ldots, X_p$ have a smooth and strictly positive density $\pi$ on $\mathbb{R}^p$. A recent result in [26] shows that conditional independence of the random variables $X_j$ and $X_k$ can be determined as follows:

$$X_j \perp\!\!\!\perp X_k \mid \boldsymbol{X}_{V \setminus \{jk\}} \iff \partial_{jk} \log \pi(\boldsymbol{x}) = 0, \ \forall \, \boldsymbol{x} \in \mathbb{R}^p, \tag{1}$$

where $\partial_{jk}(\cdot)$ denotes the $jk$-th mixed partial derivative. Here, we define the *generalized precision* as the matrix $\Omega$, where $\Omega_{jk} = \mathbb{E}_\pi \left[ |\partial_{jk} \log \pi(\boldsymbol{x})| \right]$. Note that $\Omega_{jk} = 0$ is a sufficient condition that variables $X_j$ and $X_k$ be conditionally independent. Thus, finding the zeros in the matrix $\Omega$ is equivalent to finding the graphical structure underlying the density $\pi$. Note that the zeros of the precision matrix in a Gaussian setting encode the same information—the lack of an edge—as the zeros in the generalized precision introduced here.

Our approach identifies the zeros of $\Omega$ and thus the edge set $E$ in an iterative fashion via the algorithm SING, outlined in §4. Note that computation of an entry of the generalized precision relies on an expression for the density $\pi$. We represent $\pi$ and also estimate $\Omega$ using the notion of a *transport map* between probability distributions. This map is estimated from independent samples $\boldsymbol{x}^{(i)} \sim \pi, i = 1, \ldots, n$, as described in §3. Using a map, of the particular form described below, offers several advantages: (1) computing the generalized precision given the map is efficient (e.g., the result of a convex optimization problem); (2) the transport map itself enjoys a notion of sparsity that directly relates to the Markov structure of the data-generating distribution; (3) a coarse map may capture these Markov properties without perfectly estimating the high-dimensional density $\pi$.

Let us first summarize our objective and proposed approach. We aim to solve the following graph recovery problem:

> Given samples $\{\boldsymbol{x}^{(i)}\}_{i=1}^n$ from some unknown distribution, find the dependence graph of this distribution and the associated Markov properties.

Our proposed approach loosely follows these steps:

- Estimate a transport map from samples
- Given an estimate of the map, compute the generalized precision $\Omega$
- Threshold $\Omega$ to identify a (sparse) graph
- Given a candidate graphical structure, re-estimate the map. Iterate...

The final step—re-estimating the map, given a candidate graphical structure—makes use of a strong connection between the sparsity of the graph and the sparsity of the transport map (as shown by [26] and described in §3.2). Sparsity in the graph allows for sparsity in the map, and a sparser map allows for improved estimates of $\Omega$. This is the motivation behind an iterative algorithm.

## 3 Transport maps

The first step of SING is to estimate a transport map from samples [13]. A transport map induces a deterministic coupling of two probability distributions [22, 15, 18, 26]. Here, we build a map between the distribution generating the samples (i.e., $\boldsymbol{X} \sim \pi$) and a standard normal distribution $\eta = \mathcal{N}(0, I_p)$. As described in [28, 2], given two distributions with smooth and strictly positive densities $(\pi, \eta)$,[1] there exists a monotone map $S : \mathbb{R}^p \to \mathbb{R}^p$ such that $S_\sharp \pi = \eta$ and $S^\sharp \eta = \pi$, where

$$S_\sharp \pi(\boldsymbol{y}) = \pi \circ S^{-1}(\boldsymbol{y}) \det \left( \nabla S^{-1}(\boldsymbol{y}) \right) \tag{2}$$

$$S^\sharp \eta(\boldsymbol{x}) = \eta \circ S(\boldsymbol{x}) \det \left( \nabla S(\boldsymbol{x}) \right). \tag{3}$$

We say $\eta$ is the *pushforward* density of $\pi$ by the map $S$, and similarly, $\pi$ is the *pullback* of $\eta$ by $S$. Many possible transport maps satisfy the measure transformation conditions above. In this work, we restrict our attention to lower triangular monotone increasing maps. [22, 7, 2] show that, under the conditions above, there exists a unique lower triangular map $S$ of the form

$$S(\boldsymbol{x}) = \begin{bmatrix} S^1(x_1) \\ S^2(x_1, x_2) \\ S^3(x_1, x_2, x_3) \\ \vdots \\ S^p(x_1, \ldots\ldots, x_p) \end{bmatrix},$$

with $\partial_k S^k > 0$. The qualifier "lower triangular" refers to the property that each component of the map $S^k$ only depends on variables $x_1, \ldots, x_k$. The space of such maps is denoted $\mathcal{S}_\Delta$.

As an example, consider a normal random variable: $\boldsymbol{X} \sim \mathcal{N}(0, \Sigma)$. Taking the Cholesky decomposition of the covariance $KK^T = \Sigma$, then $K^{-1}$ is a linear operator that maps (in distribution) $\boldsymbol{X}$ to a random variable $\boldsymbol{Y} \sim \mathcal{N}(0, I_p)$, and similarly, $K$ maps $\boldsymbol{Y}$ to $\boldsymbol{X}$. In this example, we associate the map $K^{-1}$ with $S$, since it maps the more exotic distribution to the standard normal:

$$S(\boldsymbol{X}) = K^{-1}\boldsymbol{X} \stackrel{d}{=} \boldsymbol{Y}, \qquad S^{-1}(\boldsymbol{Y}) = K\boldsymbol{Y} \stackrel{d}{=} \boldsymbol{X}.$$

In general, however, the map $S$ may be nonlinear. This is exactly what allows us to represent and capture arbitrary non-Gaussianity in the samples. The monotonicity of each component of the map (that is, $\partial_k S^k > 0$) can be enforced by using the following parameterization:

$$S^k(x_1, \ldots, x_k) = c_k(x_1, \ldots, x_{k-1}) + \int_0^{x_k} \exp\{h_k(x_1, \ldots, x_{k-1}, t)\} dt,$$

with functions $c_k : \mathbb{R}^{k-1} \to \mathbb{R}$ and $h_k : \mathbb{R}^k \to \mathbb{R}$. Next, a particular form for $c_k$ and $h_k$ is specified; in this work, we use a linear expansion with Hermite polynomials for $c_k$ and Hermite functions for $h_k$. An important choice is the maximum degree $\beta$ of the polynomials. With higher degree, the computational difficulty of the algorithm increases by requiring the estimation of more coefficients in the expansion. This trade-off between higher degree (which captures more possible nonlinearity) and computational expense is a topic of current research [1]. The space of lower triangular maps, parameterized in this way, is denoted $\mathcal{S}_\Delta^\beta$. Computations in the transport map framework are performed using the software TransportMaps [27].

### 3.1 Optimization of map coefficients is an MLE problem

Let $\boldsymbol{\alpha} \in \mathbb{R}^{n_\alpha}$ be the vector of coefficients that parameterize the functions $c_k$ and $h_k$, which in turn define a particular instantiation of the transport map $S_{\boldsymbol{\alpha}} \in \mathcal{S}_\Delta^\beta$. (We include the subscript in this subsection to emphasize that the map depends on its particular parameterization, but later drop it for notational efficiency.) To complete the estimation of $S_{\boldsymbol{\alpha}}$, it remains to optimize for the coefficients $\boldsymbol{\alpha}$. This optimization is achieved by minimizing the Kullback-Leibler divergence between the density in question, $\pi$, and the pullback of the standard normal $\eta$ by the map $S_{\boldsymbol{\alpha}}$ [27]:

$$\boldsymbol{\alpha}^* = \underset{\boldsymbol{\alpha}}{\operatorname{argmin}} \, \mathcal{D}_{KL}\left(\pi || S_{\boldsymbol{\alpha}}^\sharp \eta\right) \tag{4}$$

$$= \underset{\boldsymbol{\alpha}}{\operatorname{argmin}} \, \mathbb{E}_\pi\left(\log \pi - \log S_{\boldsymbol{\alpha}}^\sharp \eta\right) \tag{5}$$

$$\approx \underset{\boldsymbol{\alpha}}{\operatorname{argmax}} \, \frac{1}{n}\sum_{i=1}^n \log\left(S_{\boldsymbol{\alpha}}^\sharp \eta\left(\boldsymbol{x}^{(i)}\right)\right) = \hat{\boldsymbol{\alpha}}. \tag{6}$$

As shown in [13, 17], for standard Gaussian $\eta$ and lower triangular $S$, this optimization problem is convex and separable across dimensions $1, \ldots, p$. Moreover, by line (6), the solution to the optimization problem is a maximum likelihood estimate $\hat{\boldsymbol{\alpha}}$. Given that the $n$ samples are random, $\hat{\boldsymbol{\alpha}}$ converges in distribution as $n \to \infty$ to a normal random variable whose mean is the exact minimizer $\boldsymbol{\alpha}^*$, and whose variance is $I^{-1}(\boldsymbol{\alpha}^*)/n$, where $I(\alpha)$ is the Fisher information matrix. That is:

$$\hat{\boldsymbol{\alpha}} \sim \mathcal{N}\left(\boldsymbol{\alpha}^*, \frac{1}{n}I^{-1}(\boldsymbol{\alpha}^*)\right), \text{ as } n \to \infty. \tag{7}$$

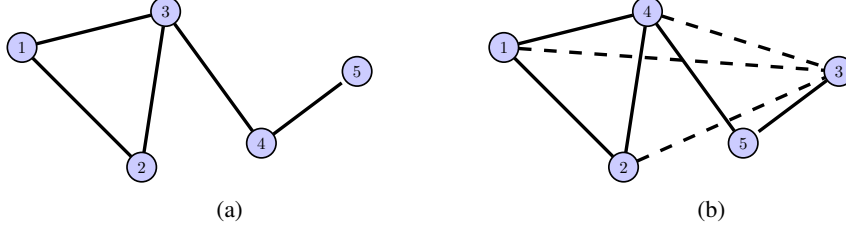

Figure 1: (a) A sparse graph with an optimal ordering; (b) Suboptimal ordering induces extra edges.

Optimizing for the map coefficients yields a representation of the density $\pi$ as $S_{\boldsymbol{\alpha}}^{\sharp}\eta$. Thus, it is now possible to compute the conditional independence scores with the generalized precision:

$$\Omega_{jk} = \mathbb{E}_{\pi}\left[|\partial_{jk}\log\pi(\boldsymbol{x})|\right] = \mathbb{E}_{\pi}\left[\left|\partial_{jk}\log S_{\boldsymbol{\alpha}}^{\sharp}\eta(\boldsymbol{x})\right|\right] \tag{8}$$

$$\approx \frac{1}{n}\sum_{i=1}^{n}\left|\partial_{jk}\log S_{\boldsymbol{\alpha}}^{\sharp}\eta\left(\boldsymbol{x}^{(i)}\right)\right| = \hat{\Omega}_{jk}. \tag{9}$$

The next step is to threshold $\hat{\Omega}$. First, however, we explain the connection between the two notions of sparsity—one of the graph and the other of the map.

### 3.2 Sparsity and ordering of the transport map

Because the transport maps are lower triangular, they are in some sense already sparse. However, it may be possible to prescribe more sparsity in the form of the map. [26] showed that the Markov structure associated with the density $\pi$ yields tight lower bounds on the sparsity pattern $\mathcal{I}_S$, where the latter is defined as the set of all pairs $(j, k), j < k$, such that the $k$th component of the map does not depend on the $j$th variable: $\mathcal{I}_S := \{(j, k) : j < k, \partial_j S^k = 0\}$. The variables associated with the complement of this set are called *active*. Moreover, these sparsity bounds can be identified by simple graph operations; see §5 in [26] for details. Essentially these operations amount to identifying the intermediate graphs produced by the variable elimination algorithm, but they do *not* involve actually performing variable elimination or marginalization. The process starts with node $p$, creates a clique between all its neighbors, and then "removes" it. The process continues in the same way with nodes $p - 1$, $p - 2$, and so on until node 1. The edges in the resulting (induced) graph determine the sparsity pattern of the map $\mathcal{I}_S$. In general, the induced graph will be more highly connected unless the original graph is chordal. Since the set of added edges, or fill-in, depends on the ordering of the nodes, it is beneficial to identify an ordering that minimizes it. For example, consider the graph in Figure 1a. The corresponding map has a nontrivial sparsity pattern, and is thus more sparse than a dense lower triangular map:

$$S(\boldsymbol{x}) = \begin{bmatrix} S^1(x_1) \\ S^2(x_1, x_2) \\ S^3(x_1, x_2, x_3) \\ S^4(\phantom{x_1, x_2,} x_3, x_4) \\ S^5(\phantom{x_1, x_2, x_3,} x_4, x_5) \end{bmatrix}, \qquad \mathcal{I}_S = \{(1, 4), (2, 4), (1, 5), (2, 5), (3, 5)\}. \tag{10}$$

Now consider Figure 1b. Because of the suboptimal ordering, edges must be added to the induced graph, shown in dashed lines. The resulting map is then less sparse than in 1a: $\mathcal{I}_S = \{(1, 5), (2, 5)\}$.

An ordering of the variables is equivalent to a permutation $\varphi$, but the problem of finding an optimal permutation is NP-hard, and so we turn to heuristics. Possible schemes include so-called min-degree and min-fill [8]. Another that we have found to be successful in practice is reverse Cholesky, i.e., the reverse of a good ordering for sparse Cholesky factorization [24]. We use this in the examples below.

The critical point here is that sparsity in the graph implies sparsity in the map. The space of maps that respect this sparsity pattern is denoted $\mathcal{S}_{\mathcal{I}}^{\beta}$. A sparser map can be described by fewer coefficients $\boldsymbol{\alpha}$, which in turn decreases their total variance when found via MLE. This improves the subsequent estimate of $\Omega$. Numerical results supporting this claim are shown in Figure 2 for a Gaussian grid graph, $p = 16$. The plots show three levels of sparsity: "under," corresponding to a dense lower

triangular map; "exact," in which the map includes only the necessary active variables; and "over," corresponding to a diagonal map. In each case, the variance decreases with increasing sample size, and the sparser the map, the lower the variance. However, non-negligible bias is incurred when the map is over-sparsified; see Figure 2b. Ideally, the algorithm would move from the under-sparsified level to the exact level.

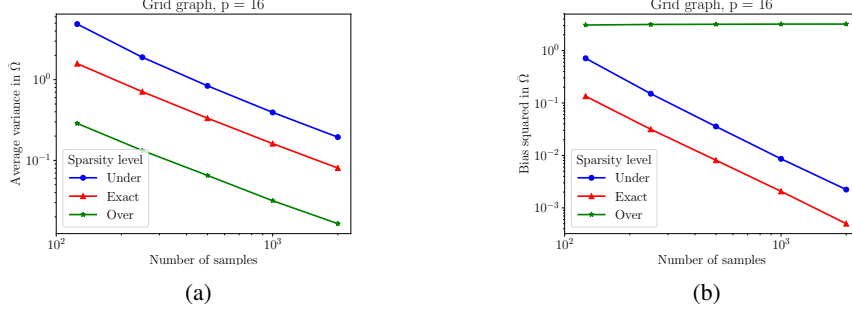

Figure 2: (a) Variance of $\hat{\Omega}_{jk}$ decreases with fewer coefficients and/or more samples; (b) Bias in $\hat{\Omega}_{jk}$ occurs with oversparsification. The bias and variance of $\hat{\Omega}$ are computed using the Frobenius norm.

# 4 Algorithm: SING

We now present the full algorithm. Note that the ending condition is controlled by a variable DECREASING, which is set to true until the size of the recovered edge set is no longer decreasing. The final ingredient is the thresholding step, explained in §4.1. Subscripts $l$ in the algorithm refer to the given quantity at that iteration.

---

**Algorithm 1:** Sparsity Identification in Non-Gaussian distributions (SING)

---

    **input** : $n$ i.i.d. samples $\{\boldsymbol{x}^{(i)}\}_{i=1}^n \sim \pi$, maximum polynomial degree $\beta$
    **output** : sparse edge set $\hat{E}$

    **define** : $\mathcal{I}_{S_1} = \{\emptyset\}$, $l = 1$, $|\hat{E}_0| = p(p-1)/2$, DECREASING = TRUE
1  **while** *DECREASING = TRUE* **do**
2     Estimate transport map $S_l \in \mathcal{S}_{\mathcal{I}_l}^\beta$, where $S_{l\sharp}\pi = \eta$
3     Compute $(\hat{\Omega}_l)_{jk} = \frac{1}{n}\sum_{i=1}^n \left| \partial_{jk} \log S_{\boldsymbol{\alpha}}^\sharp \eta \left( \boldsymbol{x}^{(i)} \right) \right|$
4     Threshold $\hat{\Omega}_l$
5     Compute $|\hat{E}_l|$ (the number of edges in the thresholded graph)
6     **if** $|\hat{E}_l| < |\hat{E}_{l-1}|$ **then**
7         Find appropriate permutation of variables $\varphi_l$ (for example, reverse Cholesky ordering)
8         Identify sparsity pattern of subsequent map $\mathcal{I}_{S_{l+1}}$
9         $l \leftarrow l + 1$
10     **else**
11         DECREASING = FALSE

---

SING is not a strictly greedy algorithm—neither for the sparsity pattern of the map nor for the edge removal of the graph. First, the process of identifying the induced graph may involve fill-in, and the extent of this fill-in might be larger than optimal due to the ordering heuristics. Second, the estimate of the generalized precision is noisy due to finite sample size, and this noise can add randomness to a thresholding decision. As a result, a variable that is set as inactive may be reactivated in subsequent iterations. However, we have found that oscillation in the set of active variables is a rare occurence. Thus, checking that the total number of edges is nondecreasing (as a global measure of sparsity) works well as a practical stopping criterion.

## 4.1 Thresholding the generalized precision

An important component of this algorithm is a thresholding of the generalized precision. Based on literature [3] and numerical results, we model the threshold as $\tau_{jk} = \delta\rho_{jk}$, where $\delta$ is a tuning parameter and $\rho_{jk} = [\mathbb{V}(\hat{\Omega}_{jk})]^{1/2}$ (where $\mathbb{V}$ denotes variance). Note that a threshold $\tau_{jk}$ is computed at each iteration and for every off-diagonal entry of $\Omega$. More motivation for this choice is given in the scaling analysis of the following section. The expression (7) yields an estimate of the variances of the map coefficients $\hat{\boldsymbol{\alpha}}$, but this uncertainty must still be propagated to the entries of $\Omega$ in order to compute $\rho_{jk}$. This is possible using the delta method [16], which states that if a sequence of one-dimensional random variables satisfies

$$\sqrt{n}\left|X^{(n)} - \theta\right| \xrightarrow{d} \mathcal{N}\left(\mu, \sigma^2\right),$$

then for a function $g(\theta)$,

$$\sqrt{n}\left|g\left(X^{(n)}\right) - g\left(\theta\right)\right| \xrightarrow{d} \mathcal{N}\left(g(\mu), \sigma^2|g'(\theta)|^2\right).$$

The MLE result also states that the coefficients are normally distributed as $n \to \infty$. Thus, generalizing this method to vector-valued random variables gives an estimate for the variance in the entries of $\Omega$, as a function of $\boldsymbol{\alpha}$, evaluated at the true minimizer $\boldsymbol{\alpha}^*$:

$$\rho_{jk}^2 \approx (\nabla_{\boldsymbol{\alpha}}\Omega_{jk})^T \left(\frac{1}{n}I^{-1}(\boldsymbol{\alpha})\right)(\nabla_{\boldsymbol{\alpha}}\Omega_{jk})\Big|_{\boldsymbol{\alpha}^*}. \tag{11}$$

## 5 Scaling analysis

In this section, we derive an estimate for the number of samples needed to recover the exact graph with some given probability. We consider a one-step version of the algorithm, or in other words: what is the probability that the correct graph will be returned after a single step of SING? We also assume a particular instantiation of the transport map, and that $\kappa$, the minimum non-zero edge weight in the true generalized precision, is given. That is, $\kappa = \min_{j \neq k, \Omega_{jk} \neq 0}(\Omega_{jk})$.

There are two possibilities for each pair $(j, k)$, $j < k$: the edge $e_{jk}$ does exist in the true edge set $E$ (case 1), or it does not (case 2). In case 1, the estimated value should be greater than its variance, up to some level of confidence, reflected in the choice of $\delta$: $\Omega_{jk} > \delta\rho_{jk}$. In the worst case, $\Omega_{jk} = \kappa$, so it must be that $\kappa > \delta\rho_{jk}$. On the other hand, in case 2, in which the edge does not exist, then similarly $\kappa - \delta\rho_{jk} > 0$.

If $\rho_{jk} < \kappa/\delta$, then by equation (11), we have

$$\frac{1}{n}\left(\nabla_{\boldsymbol{\alpha}}\Omega_{jk}\right)^T I^{-1}(\boldsymbol{\alpha})\left(\nabla_{\boldsymbol{\alpha}}\Omega_{jk}\right) < \left(\frac{\kappa}{\delta}\right)^2 \tag{12}$$

and so it must be that the number of samples

$$n > \left(\nabla_{\boldsymbol{\alpha}}\Omega_{jk}\right)^T I^{-1}(\boldsymbol{\alpha})\left(\nabla_{\boldsymbol{\alpha}}\Omega_{jk}\right)\left(\frac{\delta}{\kappa}\right)^2. \tag{13}$$

Let us define the RHS above as $n_{jk}^*$ and set $n^* = \max_{j \neq k}\left(n_{jk}^*\right)$.

Recall that the estimate in line (9) contains the absolute value of a normally distributed quantity, known as a folded normal distribution. In case 1, the mean is bounded away from zero, and with small enough variance, the folded part of this distribution is negligible. In case 2, the mean (before taking the absolute value) is zero, and so this estimate takes the form of a half-normal distribution.

Let us now relate the level of confidence as reflected in $\delta$ to the probability $z$ that an edge is correctly estimated. We define a function for the standard normal (in case 1) $\phi_1 : \mathbb{R}^+ \to (0, 1)$ such that $\phi_1(\delta_1) = z_1$ and its inverse $\delta_1 = \phi_1^{-1}(z_1)$, and similarly for the half-normal with $\phi_2$, $\delta_2$, and $z_2$. Consider the event $B_{jk}$ as the event that edge $e_{jk}$ is estimated incorrectly:

$$B_{jk} = \left\{\left((e_{jk} \in E) \cap (\hat{e}_{jk} \notin \hat{E})\right) \cup \left((e_{jk} \notin E) \cap (\hat{e}_{jk} \in \hat{E})\right)\right\}.$$

In case 1,

$$\delta_1 \rho_{jk} < \kappa \implies P(B_{jk}) < \frac{1}{2}(1 - z_1)$$

where the factor of $1/2$ appears because this event only occurs when the estimate is below $\kappa$ (and not when the estimate is high). In case 2, we have

$$\delta_2 \rho_{jk} < \kappa \implies P(B_{jk}) < (1 - z_2).$$

To unify these two cases, let us define $z$ where $1 - z = (1 - z_1)/2$, and set $z = z_2$. Finally, we have $(B_{jk}) < (1 - z)$, $j < k$.

Now we bound the probability that at least one edge is incorrect with a union bound:

$$P\left(\bigcup_{j<k} B_{jk}\right) \leq \sum_{j<k} P(B_{jk}) \tag{14}$$

$$= \frac{1}{2}p(p-1)(1-z). \tag{15}$$

Note $p(p-1)/2$ is the number of possible edges. The probability that an edge is incorrect increases as $p$ increases, and decreases as $z$ approaches 1. Next, we bound this probability of recovering an incorrect graph by $m$. Then $p(p-1)(1-z) < 2m$ which yields $z > 1 - 2m/(p(p-1))$. Let

$$\delta^* = \max[\delta_1, \delta_2] = \max\left[\phi_1^{-1}\left(1 - \frac{2m}{p(p-1)}\right), \phi_2^{-1}\left(1 - \frac{2m}{p(p-1)}\right)\right]. \tag{16}$$

Therefore, to recover the correct graph with probability $m$ we need at least $n^*$ samples, where

$$n^* = \max_{j \neq k} \left\{ (\nabla_{\boldsymbol{\alpha}} \Omega_{jk})^T I^{-1}(\boldsymbol{\alpha}) (\nabla_{\boldsymbol{\alpha}} \Omega_{jk}) \left(\frac{\delta^*}{\kappa}\right)^2 \right\}.$$

## 6 Examples

### 6.1 Modified Rademacher

Consider $r$ pairs of random variables $(X, Y)$, where:

$$X \sim \mathcal{N}(0, 1) \tag{17}$$
$$Y = WX, \quad \text{with } W \sim \mathcal{N}(0, 1). \tag{18}$$

(A common example illustrating that two random variables can be uncorrelated but not independent uses draws for $W$ from a Rademacher distribution, which are $-1$ and $1$ with equal probability.) When $r = 5$, the corresponding graphical model and support of the generalized precision are shown in Figure 3. The same figure also shows the one- and two-dimensional marginal distributions for one pair $(X, Y)$. Each 1-dimensional marginal is symmetric and unimodal, but the two-dimensional marginal is quite non-Gaussian.

Figures 4a–4c show the progression of the identified graph over the iterations of the algorithm, with $n = 2000$, $\delta = 2$, and maximum degree $\beta = 2$. The variables are initially permuted to demonstrate that the algorithm is able to find a good ordering. After the first iteration, one extra edge remains. After the second, the erroneous edge is removed and the graph is correct. After the third, the sparsity of the graph has not changed and the recovered graph is returned as is. Importantly, an assumption of normality on the data returns the incorrect graph, displayed in Figure 4d. (This assumption can be enforced by using a linear transport map, or $\beta = 1$.) In fact, not only is the graph incorrect, the use of a linear map fails to detect any edges at all and deems the ten variables to be independent.

### 6.2 Stochastic volatility

As a second example, we consider data generated from a stochastic volatility model of a financial asset [23, 6]. The log-volatility of the asset is modeled as an autoregressive process at times $t = 1, \ldots, T$. In particular, the state at time $t + 1$ is given as

$$Z_{t+1} = \mu + \phi(Z_t - \mu) + \epsilon_t, \quad \epsilon_t \sim \mathcal{N}(0, 1) \tag{19}$$

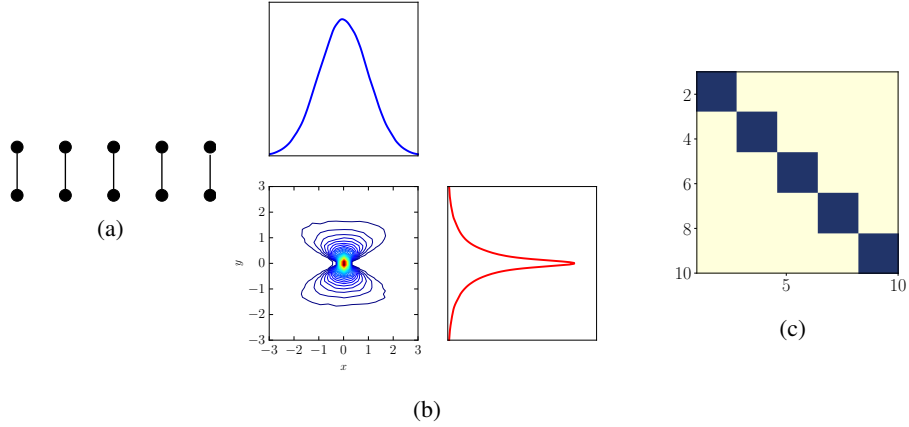

(a)

(b)

(c)

Figure 3: (a) The undirected graphical model; (b) One- and two-dimensional marginal distributions for one pair $(X, Y)$; (c) Adjacency matrix of true graph (dark blue corresponds to an edge, off-white to no edge).

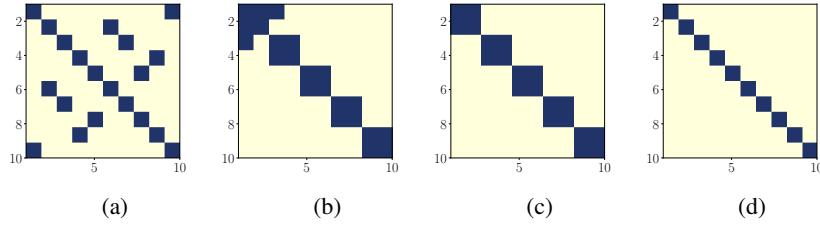

(a)                    (b)                    (c)                    (d)

Figure 4: (a) Adjacency matrix of original graph under random variable permutation; (b) Iteration 1; (c) Iterations 2 and 3 are identical: correct graph recovered via SING with $\beta = 2$; (d) Recovered graph, using SING with $\beta = 1$.

where

$$Z_0 | \mu, \phi \sim \mathcal{N}\left(\mu, \frac{1}{1 - \phi^2}\right), \quad \mu \sim \mathcal{N}(0, 1) \tag{20}$$

$$\phi = 2 \frac{e^{\phi^*}}{1 + e^{\phi^*}} - 1, \quad \phi^* \sim \mathcal{N}(3, 1). \tag{21}$$

The corresponding graph is depicted in Figure 6. With $T = 6$, samples were generated from the posterior distribution of the state $\boldsymbol{Z}_{1:6}$ and hyperparameters $\mu$ and $\phi$, given noisy measurements of the state. Using a relatively large number of samples $n = 15000$, $\delta = 1.5$, and $\beta = 2$, the correct graph is recovered, shown in Figure 6a. With the same amount of data, a linear map returns the incorrect graph—having both missing and spurious additional edges. The large number of samples is required

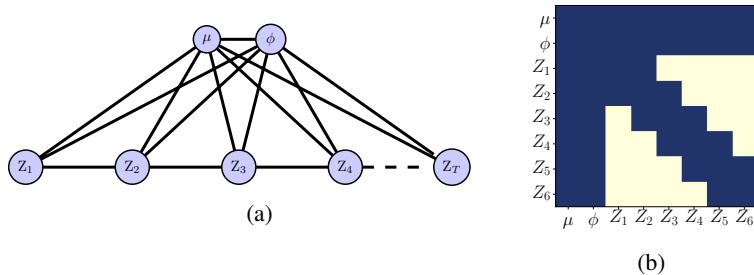

(a)

(b)

Figure 5: (a) The graph of the stochastic volatility model; (b) Adjacency matrix of true graph.

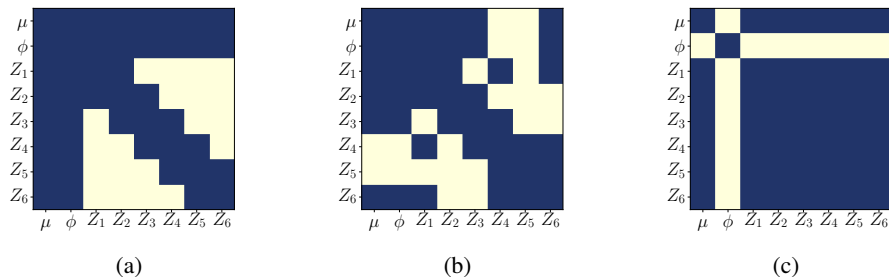

|       |       |       |
|-------|-------|-------|
| (a)   | (b)   | (c)   |

Figure 6: Recovered graphs using: (a) SING, $\beta = 2$, $n = 15000$; (b) SING, $\beta = 1$; (c) GLASSO.

because the edges between hyperparameters and state variables are quite weak. Magnitudes of the entries of the generalized precision (scaled to have maximum value 1) are displayed in Figure 7a. The stronger edges may be recovered with a much smaller number of samples ($n = 2000$), however; see Figure 7b. This example illustrates the interplay between the minimum edge weight $\kappa$ and the number of samples needed, as seen in the previous section. In some cases, it may be more reasonable to expect that, given a fixed number of samples, SING could recover edges with edge weight above some $\kappa_{\min}$, but would not reliably discover edges below that cutoff. Strong edges could also be discovered using fewer samples and a modified SING algorithm with $\ell_1$ penalties (a modification to the algorithm currently under development).

For comparison, Figure 6c shows the graph produced by assuming that the data are Gaussian and using the GLASSO algorithm [4]. Results were generated for 40 different values of the tuning parameter $\lambda \in (10^{-6}, 1)$. The result shown here was chosen such that the sparsity level is locally constant with respect to $\lambda$, specifically at $\lambda = .15$. Here we see that using a Gaussian assumption with non-Gaussian data overestimates edges among state variables and underestimates edges between state variables and the hyperparameter $\phi$.

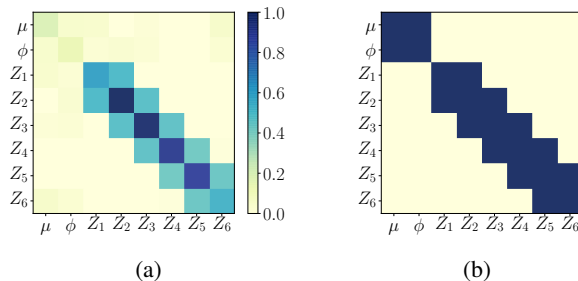

|       |       |
|-------|-------|
| (a)   | (b)   |

Figure 7: (a) The scaled generalized precision matrix $\hat{\Omega}$; (b) Strong edges recovered via SING, $n = 2000$.

## 7   Discussion

The scaling analysis presented here depends on a particular representation of the transport map. An interesting open question is: What is the information-theoretic (representation-independent) lower bound on the number of samples needed to identify edges in the non-Gaussian setting? This question relates to the notion of an *information gap*: any undirected graph satisfies the Markov properties of an infinite number of distributions, and thus identification of the graph should require less information than that of the distribution. Formalizing these notions is an important topic of future work.

**Acknowledgments**

This work has been supported in part by the AFOSR MURI on "Managing multiple information sources of multi-physics systems," program officer Jean-Luc Cambier, award FA9550-15-1-0038. We would also like to thank Daniele Bigoni for generous help with code implementation and execution.

## Footnotes

[1] Regularity assumptions on $\pi, \eta$ can be substantially relaxed, though (2) and (3) may need modification [2].

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
