[Reviews · NeurIPS 2017]

Reviewer 1



In the paper "Beyond normality: Learning sparse probabilistic graphical models in the non-Gaussian setting", the authors proposed an algorithm which estimates a sparse multivariate non-Gaussian distribution from a set of observed variables. This targeted problem is not as well studied as in Gaussian MRFs or discrete MRFs, and therefore this is an important problem (one real-world example is the cloud cover formation problem mentioned by the authors). The design of the algorithm is based on a transport map, a concept that is not well-known in the machine learning and statistics community (as far as I know). On one hand, I feel such new concept should be encouraged and welcomed by the machine learning and statistics community. On the other hand, I feel the concept can be introduced in a better way by proving more intuition of the concept. For example, the authors can provide more insight about the lower triangular monotone increasing maps (introduced at the bottom of Page 2), Hermite polynomials (for c_k) and Hermite functions (for h_k). In the meantime, I'm a little concerned that some related work (e.g. [1]) is not published yet and I wonder whether the proposed method in the current submission relies heavily on other people's unpublished work (for accuracy and validity concern). During the rebuttal phase, the authors have addressed my concern in their *Additional clarifications* section.

Reviewer 2



This paper presents a method for identifying the independence structure of undirected probabilistic graphical models with continuous but non-Gaussian distributions, using SING, a novel iterative algorithm based on transport maps. The authors derive an estimate for the number of samples needed to recover the exact underlying graph structure with some probability and demonstrate empirically that SING can indeed recover this structure on two simple domains, where comparable methods that make invalid assumptions about the underlying data-generating process fail. Quality. The paper seems technically sound, with its claims and conclusions supported by existing and provided theoretical and empirical results. The authors mostly do a good job of justifying their approach, but do not discuss potential issues with the algorithm. For example, what is the complexity of SING? Since it has a variable-elimination-like aspect, is complexity exponential in the treewidth of the generated graph? If so, how is this approach feasible for non-sparse graphs? In that vein, I would like to see how SING fares empirically when applied to more realistic and higher-dimensional problems and in comparison to other standard MRF-learning approaches. The latter typically make overly strong assumptions, but can still perform well given that there are rarely enough data points for true structures to be recovered. Clarity. The writing is good, and each subsection is clear, but the authors should better define their notation and spend more time explaining the algorithm as a whole and gluing the pieces together. In particular, what exactly happens in subsequent iterations? A quick example or a brief high-level walk-through of the algorithm would be helpful here to assemble all of the pieces that have been described into a coherent whole. Also, is the amount of sparsity that is estimated monotonic over iterations? If SING over-estimates the amount of sparsity present in the data, can it then reduce the amount of sparsity it is estimating? The sparsity pattern identified on line 8 of Alg. 1 is used in line 2, correct? (Note that you use I_{S_{i+1}} and S^B_{I_{i}}, respectively, on these lines – is I_i the same as I_{S_{i+1}} from the previous iteration?) It’s not clear to me how this sparsity pattern gets used in the estimation of the transport map. You should more clearly define where the original and induced graphs come from. From what I can tell, the original graph is the result of the thresholded \Sigma (but no graph is discussed in Alg. 1), and the induced graph comes from some variable-elimination-like algorithm, but it’s not clear how this actually happens. Does Alg. 1 perform variable elimination? What are the factors? Why do you need to do this? Originality. The approach seems novel and original, although I am not that familiar with transport maps and their corresponding literature. Significance. The presented approach seems interesting and potentially useful, but lacks sufficient evaluation as the authors do not evaluate or discuss the complexity of their algorithm and only evaluate on two simple problems. It would also be useful to properly conclude section 5 by better explaining and contextualizing the result. In particular, is the number of samples needed high or low relative to other works that estimate similar types of graph structures, or is this completely novel analysis? Overall. With additional results, editing, and a more cohesive explanation of their approach, this could be a successful publication, but I do not think it has reached that threshold as it stands. Detailed comments: - Please define your notation. For example, in the introduction you should define the independence symbol and \bf{x}_{jk}. In section 2, you use both d_j d_k for (I think) partial derivatives, but then later use d_{jk} for (I assume) the same quantity, but define neither. You should define det() and the circle, which is typically used for function composition. The V on line 154 should be defined. The projection (?) operator on the equation below line 162 should be defined. Etc. - Line 75: inline citations should be Author (Year), instead of the number (use \citet instead of \citep if using natbib) - There are a few typos and duplicated words (e.g., lines 86, 191). - The font sizes on your figures are barely legible when printed – these should be increased. - Alg 1., line 1, “DECREASING = TRUE” does not show when printed, perhaps due to font choice or embedding. - Lines 156 and 173: These should be referred to as Equation (6) and Equation (7), not Line (6) and (7). - Figure 3d shows SING(B=1), but the B=1 case has not been explained before this, so it looks like a failure case of the algorithm. Better would be to put the success case in Figure (3d) and then show the failure case in Figure (4) (separate from the others) and discuss the effect of B=1 there. - Figure 5: it’s confusing to show the true graph (5a) as the ‘recovered’ graph. You should show the recovered adjacency matrix for SING as an added sub-figure of figure 5. Further, the x- and y- axis labels are just numbers, meaning that it’s not clear which numbers correspond to the hyperparameters, so lines 232-233 are confusing. - Figures 3-6: would be useful to have a legend or explanation somewhere that blue means an edge and white means no edge. ---- After reading the author response, I have updated my score to 'weak accept' as I agree that the NIPS community will benefit from this work and my main concerns of complexity with respect to the use of variable elimination have been satisfied. However, the exact use of the variable elimination heuristics still remains somewhat unclear to me and the authors should clarify and be more explicit about how it is used in their algorithm.

Reviewer 3



Review of "Beyond normality. Learning sparse probabilistic graphical models in the non-Gaussian setting" The paper describes a new approach for structure estimation in continuous-valued non-Gaussian graphical models using "transport maps". This appears to be an very interesting higher-order generalization of copulas -- representing a joint density of interest as a multivariate (not coordinatwise) transformation of a jointly Gaussian density. The authors argue that transport maps have sparsity properties corresponding to the Markov graph of the graphical model, which allows efficient estimation and inference. The authors also use a new measure of conditional independence based on a generalization of the information matrix applicable to non-Gaussian models. I think this is a very exciting paper, providing a new (at least for the machine learning community) approach to deal with high-dimensional non-Gaussian undirected graphical models. I think that while there are many challenges to make the approach practical -- e.g. learning high-dimensional multivariate polynomials from samples will be hard to scale to much more than 10 variables considered in the paper -- but the paper opens up a new direction of research. Perhaps some form of approximate pairwise polynomial expansion (or other parametereizations) may be fast enough to learn non-trivial models. Also it appears that the authors refer to a nontrivial amount of existing work -- but this literature is new to me -- and as far as I know it has not been presented to the ML community -- appearing in physics and geoscience journals and the like. The paper is well written, and easy to follow. The key question is the scalability of the Hermite polynomial representation of the multivariate transport maps. Is there any hope that useful approximations may be obtained using pairwise expansions (it would already be a generalization of the univariate copula transformations)? Detailed comments: 1) notation x_{-jk} needs to be explained (or perhaps use x_{\jk} which is more common to refer to conditioning on all other variables in the graph. 2) Sec. 1. page 1. "algorithms are consistent for sparse recover with few samples"... It's worth being more precise -- what do you mean by "few samples"? 3) Sparse l1-penalized logistic regression is also used in the discrete setting (in addition to [12]). "HIGH-DIMENSIONAL ISING MODEL SELECTION USING l1-REGULARIZED LOGISTIC REGRESSION", Ravikumar, Wainwright, Lafferty, 2010. 4) Cloud cover formation: what is the data? Satellite images? 5) For your definition of Omega_jk( x ) -- is there a notion of scale -- to be able to argue about "small" or "large" values? Is it possible to normalize it to be [-1,1] -- similar to correlation coefficients? Does this matrix have any properties -- e.g. positive-definiteness or cpd? 6) There are principled thresholding based approaches for estimating sparse inverse covariance matrices, e.g. "Elementary Estimators for Sparse Covariance Matrices and other Structured Moments", Yang, Lozano, Ravikumar, ICML 2014. 7) In (2) and (3) -- what conditions are there on the two maps? Is it fully general (or needs smoothness and strict positivity)? 8) The monotonicity of __the__ each component (line 86)... drop __the__... 9) I am assuming you use multivariate Hermite polynomial expansions? What degree do you consider (are you able to handle) in practice -- there's a combinatorial explosion of multivariate terms that appears daunting. How many multivariate basis functions do you use -- how much time does it take for the computational examples considered in the paper? 10) Please give the intuition why the problem is convex in a few words? Do you only need second-order moments of pi, which depend linearly on the hermite coefficients? 11) Sec 3.2 The lower bounds from the markov structure to the sparsity of the transport maps -- is this not an exact match due to fill-in in variable elimination? 12) In figure 2 (b) -- why is bias non-zero -- but depends on the number of samples? How do you define and measure bias? 13) In algorithm 1 -- does the sparsity profile of the transport map increase monotonically with iterations? Can it oscillate? Are there any guarantees / empirical observations? 14) What is the delta method? reference? 15) We consider the case in which the sparse graph is returned after a single iteration... ? How can you assume this, and how often does it happen in practice? Perhaps it's better to say that you're analyzing convergence after identifying the correct graph structure? 16) Figure 3 (b) -- very small labels. 17) What it SING with l1-penalties? Is there a reference?